# The Role of Immediate Post-Procedural Cone-Beam Computed Tomography (CBCT) in Predicting the Early Radiologic Response of Hepatocellular Carcinoma (HCC) Nodules to Drug-Eluting Bead Transarterial Chemoembolization (DEB-TACE)

**DOI:** 10.3390/jcm11237089

**Published:** 2022-11-29

**Authors:** Marco Fronda, Francesco Mistretta, Marco Calandri, Fernanda Ciferri, Floriana Nardelli, Laura Bergamasco, Paolo Fonio, Andrea Doriguzzi Breatta

**Affiliations:** 1Radiology Unit, Department of Diagnostic Imaging and Interventional Radiology, A.O.U. Città della Salute e della Scienza di Torino, Via Genova 3, 10126 Turin, Italy; 2Radiology Unit, Department of Surgical Sciences, A.O.U. Città della Salute e della Scienza di Torino, University of Torino, Via Genova 3, 10126 Turin, Italy; 3Department of Surgical Sciences, A.O.U. Città della Salute e della Scienza di Torino, University of Torino, C.so Bramante 88, 10126 Turin, Italy

**Keywords:** cone-beam computed tomography, hepatocellular carcinoma (HCC), DEB-TACE, transarterial chemoembolization

## Abstract

The purpose of this study was to evaluate the efficacy of unenhanced cone-beam computed tomography (CBCT) performed at the end of drug-eluting bead transarterial chemoembolization (DEB-TACE) in predicting HCC nodules’ early radiologic response to treatment, assessed using mRECIST criteria with a 30–60 day four-phase contrast-enhanced CT follow-up. Fifty-nine patients (81 lesions) subjected to DEB-TACE as exclusive treatment for HCC lesions (naive/relapse) between February 2020 and October 2021 were prospectively enrolled. In a post-interventional unenhanced CBCT procedure, two experienced radiologists evaluated for each lesion the overall intensity of the contrast media deposit, the homogeneity of the enhancement, and the presence of smooth and complete margins. The univariate analysis found that lesions with complete response (CR+) had a significantly higher incidence of clear and complete margins than CR− lesions (76.9% vs. 17.2%, *p* = 0.003) and a higher intensity score (67.3% vs. 27.6%, *p* = 0.0009). A Dmax <30 mm was significantly more common among CR+ than CR− lesions (92.3% vs. 69%, *p* = 0.01). These features were confirmed as significant predictors for CR+ by multivariate binary logistic regression. The homogeneity of the enhancement did not affect the DEB-TACE outcome. Post-interventional unenhanced CBCT is effective in predicting early radiological response to DEB-TACE, since the presence of an intense contrast media deposit with clear and complete margins in treated HCC lesions is associated with CR.

## 1. Introduction

Hepatocellular carcinoma (HCC) is the most frequent primary liver cancer and is predominantly associated with chronic liver disease. The main risk factors for HCC are chronic viral hepatitis (hepatitis C virus (HCV) or hepatitis B virus (HBV)), heavy alcohol consumption, diabetes, and nonalcoholic fatty liver disease (NAFLD) [1].

For the evaluation of HCC patients, the Barcelona Clinic Liver Cancer (BCLC) staging system is commonly used. According to this system, transarterial chemoembolization (TACE) is the first-line therapy for the treatment of intermediate-stage HCC, i.e., patients with multinodular unresectable HCC without vascular or extrahepatic invasion and with preserved liver function [2]. Furthermore, it is estimated that up to 40% of TACE procedures are performed in the early (BCLC-A) or, rarely, advanced (BCLC-C) stages [3,4], particularly as a bridge to liver transplantation or in a stage migration strategy [5,6].

During TACE treatment, cone-beam computed tomography (CBCT) can be useful in identifying liver lesions and their arterial feeders, improving the technical success of the treatment compared to TACE performed using DSA alone [7,8,9].

The early assessment of treatment efficacy is important for identifying suboptimal treatments that could lead to tumor progression or recurrence, and a radiologic complete response (CR) after the first TACE session is a robust predictor of a favorable outcome [10,11]. Generally, an MRI and/or CT exam is performed 1 to 3 months after treatment to evaluate tumor response, indicated by the absence of viable tumor tissue with contrast uptake [12]. CBCT could be a useful tool for assessing the efficacy of TACE in the intra-operative setting. In particular, it has been demonstrated that the complete filling of the tumor with iodized oil is predictive of a CR in conventional TACE (the intra-arterial injection of cytotoxic drugs and lipiodol, followed by the injection of embolic agents) [13,14,15]. As regards drug-eluting bead (DEB)-TACE, only a few studies have focused on the efficacy of intraprocedural CBCT in predicting treatment outcomes [16,17,18].

The aim of this study was to evaluate the efficacy of unenhanced CBCT performed at the end of TACE in predicting the early radiologic response of HCC nodules to treatment. A comparison in terms of efficacy was also conducted between the population subjected to chemoembolization with DEBs alone and the population treated with DEBs and other embolic agents.

## 2. Materials and Methods

### 2.1. Study Design

This was a single-center prospective observational study conducted on patients subjected to DEB-TACE as exclusive treatment for HCC lesions (naive or relapse) between February 2020 and October 2021.

In all patients, the decision to treat was taken by a multidisciplinary committee. Inclusion criteria were: (i) HCC diagnosed by pathologic assessment or non-invasive diagnosis criteria according to EASL-EORTC Clinical Practice Guidelines [19]; (ii) intermediate-stage HCC according to the BCLC staging system [2]; (iii) early-stage HCC in patients unsuitable for ablation, resection, or transplantation; (iv) Child–Pugh score ≤ B7; (v) Eastern Cooperative Oncology Group (ECOG) performance status 0.

The exclusion criteria were: (i) HCC lesions treated with DEB-TACE combined with radiofrequency or microwave ablation; (ii) patients in whom CBCT could not be performed (i.e., extremely obese, uncooperative, or unable to hold breath). Fifty-nine patients, with a total number of 81 lesions, satisfied all inclusion and exclusion criteria and were thus enrolled in the study.

The local ethical committee authorized the study, which was conducted in accordance with the declaration of Helsinki and national legislation. Before the TACE session, all patients were informed about the possible use of their data for study purposes. Patients’ information was anonymized before the analysis.

### 2.2. DEB-TACE Procedure

Under local anesthesia, a 5-French (F) sheath (Radifocus Introducer II, Terumo, Tokyo, Japan) was inserted in the right or left common femoral artery. The celiac axis and, if deemed necessary based on the preprocedural CT findings, the superior mesenteric artery were catheterized with a 5-F catheter (Shepherd Hook II, TEMPO SHK1.0, Cordis, Hialeah, FL, USA). An initial digital subtraction angiography (DSA) was performed with 20 mL of iopromide (Ultravist 370, Bayer Pharma, Berlin, Germany) at a flow rate of 4 mL/s. Then, the segmental or subsegmental feeding arteries of the hypervascular lesions identified were catheterized using a 1.9-F microcatheter (Progreat Lambda, Terumo, Tokyo, Japan or Carnelian, Tokai Medical Product, Aichi, Japan) with a 0.016 in or 0.014 in. microwire (Fathom, Boston Scientifics, Marlborough, MA, USA). Procedures were carried out with LifePearl 100 ± 25 μm (Terumo, Tokyo, Japan) (*n* = 75); Embozene TANDEM 100 μm (Varian, Palo Alto, CA, USA) (*n* = 2); or Hydropearl 200 ± 75 μm (Terumo, Tokyo, Japan) (*n* = 4). Bead loading was performed with 75 mg of epirubicin (Farmorubicin, Pfizer, New York, NY, USA). Once the microcatheter was in place, beads were diluted in 15 mL of contrast and saline mixed solution (50:50) and then administered under fluoroscopic guidance, in order to prevent reflux. The amount of the administered dose was adapted to reach the complete devascularization of the target lesion and the near stasis of the feeding vessel(s). In 34 cases, a second embolic agent was administered (Hydropearl 400 ± 75 μm (*n* = 20); gelatin sponge (Cutanplast, Mascia Brunelli S.p.A., Milan, Italy) (*n* = 12); or Lipiodol (Guerbet, Villepinte, France) (*n* = 2)). In 11 cases of persisting arterial flux in the target area and subcapsular nodules, near stasis was achieved with a gelatin sponge (Cutanplast, Mascia Brunelli S.p.A., Milan, Italy). A comparison was conducted in terms of efficacy between the population submitted to DEBs alone and those treated with DEBs and other embolic agents.

### 2.3. CBCT Image Protocol

All procedures were performed in a dedicated angio suite equipped with a ceiling-mounted angiographic C-arm system with a 40 × 30 cm flat panel detector (Allura Xper FD20, Philips Healthcare, Amsterdam, The Netherlands). Unenhanced CBCT was performed at the end of TACE, with the C-arm at the headend of table and the liver symmetrically in the isocenter of the C-arm rotation (from −120° to +120° in about 4 s). The datasets were automatically transferred to a dedicated workstation for reconstruction and analysis.

### 2.4. Analysis of CBCT Imaging

Two experienced radiologists, blinded to the patient’s outcome, visually assessed the overall intensity of the contrast media deposit in the HCC lesion in the unenhanced immediate post-interventional CBCT. Furthermore, the homogeneity of the enhancement and the presence of smooth and complete margins were assessed for each lesion. Analyses were performed using the score shown in Figure 1. When radiologists assigned different scores, a face-to-face review was performed to reach a consensus. The interobserver agreement was evaluated for all the analysis parameters.

### 2.5. Preoperative and Follow-Up Imaging

All patients underwent four-phase contrast-enhanced CT (CECT) within 2 months before the procedure and between 30 and 60 days after the procedure, according to the American Association for the Study of Liver Diseases (AASLD) guidelines [18]. In the preoperative CECT, the maximum diameter (Dmax) of the lesions was measured.

The study was carried out in a tertiary referral center; preoperative imaging was performed partly by our diagnostic radiology service and partly by other centers. All cases were discussed by the multidisciplinary board of the referral institution, which included at least two radiologists and one interventional radiologist.

As regards the follow-up imaging, responses were assessed on hepatic-arterial CT images using the modified Response Evaluation Criteria In Solid Tumors (mRECIST) system [12], classifying them as complete response (CR), partial response (PR), stable disease (SD), or progressive disease (PD). The FU was evaluated by two independent radiologists, blinded to the CBCT features of the treated HCC. Disagreements in interpretation were resolved by discussions to reach a consensus.

### 2.6. Statistical Analysis

Continuous variables that satisfied the Shapiro–Wilks W test for normality were expressed as average and standard deviation, and categorical variables were expressed as counts and percentages. The univariate analysis used non-parametric tests: the Mann–Whitney test for independent continuous variables and Fisher’s exact test for dichotomic variables. Multivariate analysis used binary logistic regression (BLR).

The discriminatory ability of the continuous variables with significant differences according to univariate analysis, confirmed by BLR, was measured by the area under the curve (AUC) of the receiver operating characteristic (ROC) curve, i.e., the plot of sensitivity versus (1-specificity): AUC = 0.5 corresponding to no discrimination (chance), and then increasing from 0.6 (poor) to 1 (excellent).

Significant association was indicated by *p* < 0.05 and 95% odds ratio CI, not including 1 (significant risk above 1, protection below 1). The test power for determining the absence of significant differences was set at *p* > 80%.

Analyses were performed using Statplus for Macintosh Build 8.0.1.0/Core v7.7.11, 2021 (AnalystSoft, Walnut, CA, USA).

## 3. Results

The study enrolled 59 patients, 42 of whom had one lesion, 12 two lesions, and 5 three lesions, for a total of 81 lesions (1.45 per patient), treated by TACE.

The early post-TACE radiological response of the HCC lesions was evaluated between 30 and 60 days after the procedure with four-phase CECT, according to the mRECIST criteria [12]. This revealed 52 (64.2%) CR, 18 (22.2%) PR, 9 (11.1%) SD, and 2 (2.5%) PD. For the analyses, the early radiological response of each lesion was dichotomized into CR versus others (PR, SD, and PD). Table 1 compares the variables of the 52 lesions with early radiological response (CR+) to those of the other 29 lesions (CR−).

The univariate analysis found that CR+ lesions had a significantly higher incidence of clear and complete margins than CR− lesions (76.9% vs. 17.2%, *p* = 0.003) and a higher intensity score (67.3% vs. 27.6% of lesions with an intensity score of 2, *p* = 0.0009) for the contrast media deposit according to unenhanced post-interventional CBCT.

Considering a cut-off value of 30 mm for the maximum diameter, we found that a Dmax < 30 mm was significantly more common among the CR+ than the CR− lesions (92.3% vs. 69%, *p* = 0.01).

The DEB-TACE outcome was not influenced by the homogeneity of the contrast media deposit.

There were no statistically significant differences in DEB-TACE outcome between patients treated with DEBs alone and patients treated with DEBs and other embolic agents (70.2% vs. 55.9% CR+, respectively; *p* = 0.23).

The multivariate binary logistic regression confirmed that the presence of clear and complete margins (*p* < 0.0001, OR = 17 (5−55)); a high intensity score (*p* = 0.001, OR = 6 (2−18)); and a Dmax < 30 mm (*p* = 0.03; OR = 6.3 (1.2−32)) were significant predictors for CR+.

The discriminating ability of the significant predictors was tested by the ROC curve procedure: the presence of clear and complete margins ranked first (AUC = 0.80, sensitivity 0.77, specificity 0.83, positive predictive value 0.89, negative predictive value 0.67), as shown in Figure 2, followed by a high intensity score (AUC = 0.74, sensitivity 0.67, specificity 0.72, positive predictive value 0.81, negative predictive value 0.56).

Although the discordant evaluations were resolved through discussions to reach a consensus, the agreement between the two radiologists was good/very good: for intensity, Cohen’s K = 0.87 (0.76−0.98); for homogeneity, Cohen’s K = 0.96 (0.88−1); for margins, Cohen’s K = 0.79 (0.65−0.94).

## 4. Discussion

The present study, performed on 81 HCC lesions treated with DEB-TACE, was aimed at investigating the efficacy of unenhanced CBCT after DEB-TACE in predicting the early radiologic response of HCC nodules to treatment. Unenhanced CBCT evaluation was performed through the qualitative assessment of the intensity and homogeneity of the contrast media deposit in the HCC lesion and the presence of clear and complete margins. The present series showed an overall CR rate of 64.2%, slightly higher than those reported in other per-lesion analyses [20,21,22] and in our previous experience [23]. The use of strictly calibrated microbeads and the routine use of intra-procedural CBCT could have contributed to this better result. Several studies have aimed to find predictive preoperative imaging biomarkers for the early CR of HCC after chemoembolization, with a particular focus on lesion size. In our study, a Dmax < 30 mm was significantly more common among CR+ compared to CR− lesions (92.3% vs. 69%, *p* = 0.01), and the multivariate binary logistic regression confirmed Dmax < 30 mm as a significant predictor for CR+. This result confirmed the previous findings of Zhang et al. [21] and Vesselle et al. [24], which indicated that a smaller lesion size is a good predictor of higher susceptibility to TACE.

Focusing on the unenhanced post-procedural CBCT features, we found that CR+ lesions showed a significantly higher incidence of clear and complete margins and a higher intensity score for the contrast media deposit (Figure 3). The presence of clear and complete margins showed the best discriminating ability, as measured by the standard ROC curve procedure (AUC = 0.80 with sensitivity 0.77 and specificity 0.83).

The homogeneity of the contrast media deposit did not influence the DEB-TACE outcome. This finding could be explained, in our opinion, by the common presence of small intralesional necrotic areas, determining inhomogeneous microsphere distribution inside the nodules.

Some studies have already shown that CBCT could be a useful tool for assessing the efficacy of TACE in the intra-operative setting. Jeon et al., Sun et al., and Wang et al. demonstrated that the complete filling of the tumor with iodized oil is predictive of a CR in conventional TACE (cTACE) [13,14,15]. Furthermore, the importance of assessing the safety margin of embolized areas using intraprocedural CBCT in cTACE has already been demonstrated by a study in which local tumor recurrence was significantly lower when a greater degree of iodized oil deposition occurred with a complete circumferential safety margin [25].

Whereas the distribution of embolic agents can be easily assessed in cTACE, according to some authors, the postprocedural assessment of the distribution of DEBs is limited and, in some cases, infeasible. Conversely, we found that unenhanced CBCT at the end of DEB-TACE usually allowed for a good assessment of DEB distribution.

In agreement with our findings, the study of Syha et al. [16] found that postinterventional unenhanced CBCT allowed the assessment of the contrast media deposit of DEBs in most treated HCC lesions (77%), with a tendency toward higher visibility in encapsulated HCC lesions. However, the association between the presence and features of the contrast deposit and CR after DEB-TACE has not been evaluated. Regarding DSM-TACE (degradable starch microsphere-TACE), a quantitative evaluation of intraprocedural CBCT was performed with predictive purposes in a study by Orlacchio et al. [26], showing that the diameter, volume, and density of lesions had high accuracy in predicting an early CR. In that study, a higher CR rate was observed in lesions in which the chemoembolization mixture retention had nearly the same diameter or volume value as that of the enhancing portion evaluated via preprocedural CT, and the density value was higher than that measured by the preprocedural CT.

To the best of our knowledge, however, this is the first study to perform a systematic qualitative evaluation of unenhanced CBCT at the end of DEB-TACE with predictive purposes.

In the present study, an evaluation of treatment efficacy at 30–60 day follow-up was performed with four-phase contrast-enhanced CT. The advantages of MRI (especially using liver-specific contrast agents) over CT in detecting and categorizing naïve HCC nodules are well-known. However, in everyday clinical practice, CT is more readily available and fairly accurate for assessing DEB-TACE radiologic responses according to mRECIST criteria. Currently, the international guidelines do not recommend MRI over CT for this purpose [27].

This study had some limitations. First, it was a retrospective single-center study. Second, even though an early radiologic CR is known to be a predictor of a favorable outcome [11], the impact of the predictive factors on survival was not assessed. Third, including non-naive lesions in the study could have been a confounding factor. However, no statistically significant differences in the presence of non-naive lesions between CR + and CR− were found in the univariate analysis (Table 1). Fourth, since 68 of the 81 treated nodules had diameters <30 mm, the applicability of these criteria to nodules > 30 mm must be demonstrated in further studies. Fifth, the population was inhomogeneous as regards the embolic agents used. In 47 cases, DEBs were used alone, while in cases where arterial flow persisted in the target area (34 cases), additional embolic agents were used. However, the use of additional embolic agents did not lead to significant differences in DEB-TACE outcomes. Sixth, the qualitative analysis was affected by subjectivity, even if the agreement between the two radiologists was good/very good for all parameters (intensity, homogeneity, and margins) and a consensus evaluation was reached in the case of discordant results. Moreover, qualitative analysis is currently the only feasible method of assessment, as there are no standardized methods for the quantitative analysis of interventional CBCT, considering the intrinsic inability of CBCT to provide a density value measured in Hounsfield units. Some studies have tried to estimate density values from CBCT images, defining an arbitrary unit correlated to CT Hounsfield units [18,26,28]. However, this approach might present some limitations and can hardly be generalized to different angiography/CBCT systems, so further studies in this field are needed.

In conclusion, this study demonstrated that unenhanced CBCT performed at the end of DEB-TACE is effective in predicting early radiological response to treatment, since the presence of an intense contrast media deposit with clear and complete margins in treated HCC lesions is associated with CR. Moreover, the homogeneity of the contrast media deposit has no role in predicting short-term CR to DEB-TACE.

In clinical practice, if confirmed among larger cohorts, these findings could help interventional oncologists to immediately assess the technical success of DEB-TACE and could represent a very useful guide for decisions to stop embolization or to search for additional minor feeding vessels, especially in cases of incomplete embolization margins.

## Figures and Tables

**Figure 1 jcm-11-07089-f001:**
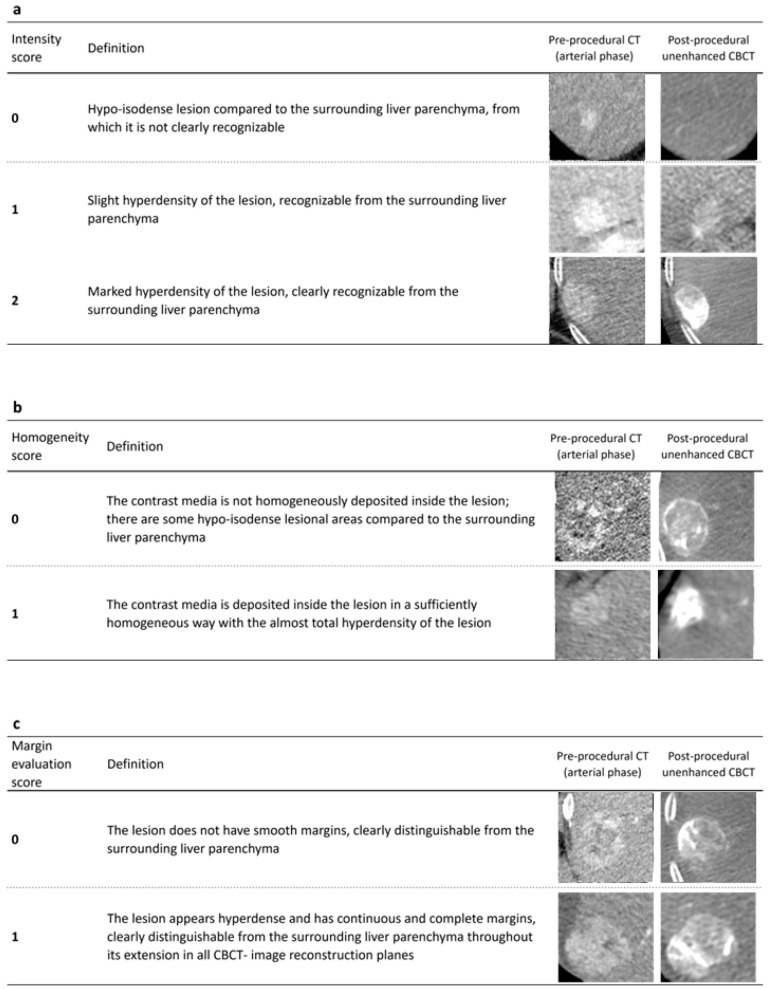
Subjective score visually assigned by the two radiologists for intensity (**a**), homogeneity (**b**), and margins (**c**) of contrast media deposit in unenhanced post-interventional CBCT.

**Figure 2 jcm-11-07089-f002:**
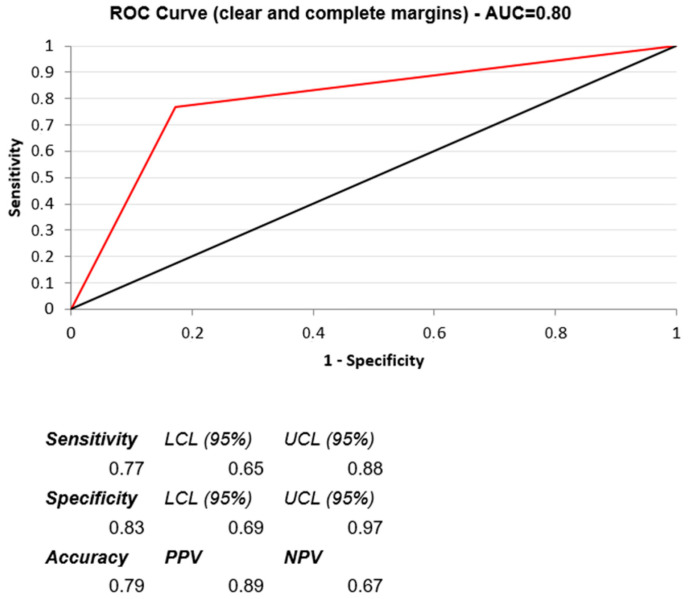
ROC curve for the presence of clear and complete margins, which showed the best predictive value for CR+: AUC = 0.80, sensitivity 0.77, specificity 0.83, PPV 0.89, and NPV 0.67.

**Figure 3 jcm-11-07089-f003:**
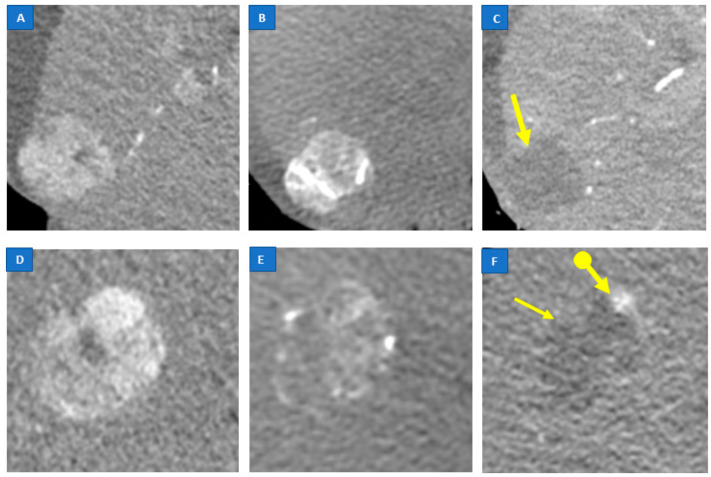
Lesion with complete response (**A**–**C**): pre-procedural CT arterial phase (**A**); unenhanced post-procedural cone-beam CT (**B**) (intensity score 2, homogeneity score 0, margin evaluation score 1); 1-month follow up CT arterial phase (**C**) showing hypodense area due to treatment without pathological viable tissue (*arrow*). Lesion with partial response (**D**–**F**): pre-procedural CT arterial phase (**D**); unenhanced post-procedural cone beam CT (**E**) (intensity score 2, homogeneity score 1, margin evaluation score 0); 1-month follow-up CT arterial phase (**F**) showing hypodense area due to treatment (*arrow*) with persistence of pathological viable tissue characterized by intense enhancement (*dot and arrow*).

**Table 1 jcm-11-07089-t001:** Univariate analysis and binary logistic regression (BLR) of demographics, clinical data, and lesion features. CR+ vs. CR− lesions.

Variable	Complete Response (CR+), *n* = 52	No Complete Response (CR−), *n* = 29	Univariate *p* Value	Multivariate BLR *p* Value	Multivariate BLR Odds Ratio (95% CI)
**Age** mean	69	70	0.44		
**Males**	42 (80.8%)	26 (89.7%)	0.37		
**Etiology**			0.715		
- HCV	26 (50%)	14 (48.3%)			
- HBV	4 (7.7%)	3 (10.3%)			
- ALD	5 (9.6%)	2 (6.9%)			
- NASH	13 (25%)	3 (10.3%)			
- Other	4 (7.7%)	7 (24.1%)			
**Naive/not naive lesions ratio**	43/9	19/10	0.11		
**Lesions with** **Dmax < 30 mm**	48 (92.3%)	20 (69%)	**0.01**	**0.03**	6.3 (1.2–32)
**Intensity score ***			**0.0009**	**0.001**	6 (2–18)
0	0	2 (6.9%)			
1	17 (32.7%)	19 (65.5%)			
2	35 (67.3%)	8 (27.6%)			
**Homogeneity score**			0.401		
0	37 (71.2%)	25 (86.2%)			
1	15 (28.8%)	4 (13.8%)			
**Margin evaluation score**			**0.003**	**<0.0001**	17 (5–55)
0	12 (23.1%)	24 (82.8%)			
1	40 (76.9%)	5 (17.2%)			
**Embolizing agents**			0.23		
- DEBs alone	33/47 (70.2%)	14/47 (29.8%)			
- DEBs with other embolizing agents	19/34 (55.9%)	15/34 (44.1%)			

* The analysis was conducted by comparing lesions with intensity score 0 or 1 with those with intensity 2. Bold *p* Values indicate significant differences.

## Data Availability

Not applicable.

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
