# Peer review of "The Role of Immediate Post-Procedural Cone-Beam Computed Tomography (CBCT) in Predicting the Early Radiologic Response of Hepatocellular Carcinoma (HCC) Nodules to Drug-Eluting Bead Transarterial Chemoembolization (DEB-TACE)"

_jcm, 2022, doi:10.3390/jcm11237089_

Round 1

Reviewer 1 Report

The Authors propose the use of immediate post-procedural Cone Beam Computed Tomography (CBCT) in predicting the early radiologic response of hepatocellular carcinoma (HCC) nodules to DEB-TACE.

Comments:

As also described by the Authors, according to the Barcelona Clinic Liver Cancer (BCLC) staging system trans-arterial chemoembolization (TACE) is the first-line therapy for the treatment of intermediate stage HCC (stage B). However, the study population is mainly represented by patients with stage A HCC, as no patient had more than 3 nodules and only 13 nodules were >30 mm in diameter. The authors should explain how the patients enrolled in the study were selected and the reasons for this discrepancy.

 Since 68 of 81 treated nodules were <30 mm, the study results could be applied to patients with up to 3 nodules with diameter <30 mm, as data on nodules >30 mm are too limited.

 The performance of intensity score and of margin evaluation score in prediction of early radiologic response are not so good. The Authors suggest, after confirmation of their results on larger series, the possibility of using this tool in “guiding the decision to stop the embolization or looking for additional minor feeding vessels, especially in case of incomplete embolization margins”. In case of additional treatment, for example, one third (12 of 36) of patients with margin evaluation score 0, CR+, could be overtreated.

 To better evaluated the utility of intensity score and of margin evaluation score n clinical practice, it could be useful:

-        To analyse the prediction of the response at 3 or 6 months after TACE

-        To analyse a combined score (intensity score + margin evaluation score)

Reviewer 2 Report

Title: The role of immediate post-procedural Cone Beam Computed Tomography
(CBCT) in predicting the early radiologic response of hepatocellular carcinoma (HCC) nodules to drug eluting beads trans-arterial chemoembolization (DEB-TACE)

General comments

The authors reported their results of Cone Beam Computed Tomography in predicting the early radiologic response of hepatocellular carcinoma (HCC) nodules to drug eluting beads trans-arterial chemoembolization (DEB-TACE). A total of 59 patients with liver cancer were included

They concluded that the post-interventional unenhanced-CBCT is effective in predicting early radiological response to DEB- TACE, since the presence of intense contrast media deposit with clear and complete margins in the treated HCC lesions is associated with CR.

Interesting topic, however on its own, the results may have little significance for clinical practice.

Major flaws in the methodology used

The methods and techniques of the study lack novelty.

Single center and single arm, with relative small sample size.

Study design: mRECIST criteria 30-60 days four-phase contrast-enhanced CT follow-up.” This is my main concern. Currently, the most appropriate imaging technique for evaluation of efficacy following the local therapy for HCC is dynamic contrast enhanced MRI, not the dynamic CT.

Minor comments

1.     Abstract: No comments.

2.     Introduction: Not refining enough and should be reduced.

3.     M & M:

—The embolic agents used in the study were different DC-beads, Lipiodol, and Gelatin sponge. It is inconsistent with the the “title”.

—MethodsPreoperative and follow-up imaging。。。 Who performed the evaluation? How?

4.     Results: No comments.

The images quality (Figure 1 and Figure 3) are very low.

Discussion: Not refining enough and should be reduced. Should accentuate the contributions in this manuscript.

Round 2

Reviewer 2 Report

No. 

Author Response

According to the comment of the second reviewer ("The embolic agents used in the study were different DC-beads, Lipiodol, and Gelatin sponge..") I suggest to add in the paper the following parts (title is ok): 1. Insert as second end-point in the purpose a separate comparison in terms of efficacy between population submitted to DC-Beads alone and DC-Beads with other embolic agents. 2. Insert in MMs section this investigation. 3. Insert in Results section this finding. 4. Add in Discussion in limitations paragraph this aspect (inhomogeneous population). 

Thanks for this suggestion. We performed an additional analysis on the embolizing agents used (DEBs alone vs DEBs with other embolic agents). The use of additional embolizing agents did not determine significant differences on DEB-TACE outcome. We inserted the analysis in the purpose of study, MMs section, Results and in Discussion. The analysis table was also updated.